# Colistin Dosing Regimens against *Pseudomonas aeruginosa* in Critically Ill Patients: An Application of Monte Carlo Simulation

**DOI:** 10.3390/antibiotics10050595

**Published:** 2021-05-17

**Authors:** Van Thi Khanh Nguyen, Preecha Montakantikul, Pramote Tragulpiankit, Jantana Houngsaitong, Mohd Fazli Shuib

**Affiliations:** 1Division of Clinical Pharmacy, Department of Pharmacy, Faculty of Pharmacy, Mahidol University, Bangkok 10400, Thailand; nguyenkhanhvan272@gmail.com (V.T.K.N.); pramote.tra@mahidol.ac.th (P.T.); jantana.hon@mahidol.ac.th (J.H.); 2Pharmacy Department, FV Hospital, Hochiminh City 70000, Vietnam; m.shuib@fvhospital.com

**Keywords:** colistin, *Pseudomonas aeruginosa*, critically ill patients, PK/PD, Monte Carlo simulation

## Abstract

Our aims are to assess various colistin dosing regimens against *Pseudomonas aeruginosa* (*P. aeruginosa*) infection in critically ill patients and to propose an appropriate regimen based on microbiological data. A Monte Carlo simulation was performed using the published colistin’s pharmacokinetic parameters of critically ill patients, the published pharmacodynamic target from a mouse thigh infection model, and the minimum inhibitory concentration (MIC) results from a Vietnamese hospital. The probability of target attainment (PTA) of 80% and cumulative fraction of response (CFR) of 90% were used to evaluate the efficacy of each regimen. Of 121 *P. aeruginosa* laboratory datasets, the carbapenem-resistant *P. aeruginosa* (CRPA) and the colistin-resistant *P. aeruginosa* rates were 29.8% and 0.8%, respectively. MIC_50,90_ were both 0.5 mg/L. The simulated results showed that at MIC of 2 mg/L, most regimens could not reach the PTA target, particularly in patients with normal renal function (Creatinine clearance (CrCl) ≥ 80 mL/min). At MIC of 0.5 mg/L and 1 mg/L, current recommendations still worked well. On the basis of these results, aside from lung infection, our study recommends three regimens against *P. aeruginosa* infection at MIC of 0.5 mg/L, 1 mg/L, and 2 mg/L. In conclusion, higher total daily doses and fractionated colistin dosing regimens could be the strategy for difficult-to-acquire PTA cases, while a less aggressive dose might be appropriate for empirical treatment in settings with low MIC_50/90_.

## 1. Introduction

*Pseudomonas aeruginosa* (*P. aeruginosa*) is one of the most concerning pathogens related to nosocomial infection [1]. Due to the emergence of multi-drug resistant (MDR) and carbapenem-resistant *P. aeruginosa* (CRPA) isolates, some serious consequences can be named such as reducing treatment of choices, inappropriate initial therapy, and delayed treatment [2,3]. The eventual outcomes are a higher mortality rate, a higher readmission rate, as well as the increase of the length of stay and treatment cost [4,5,6,7]. Given that fact, MDR *P. aeruginosa* has made it to the list of CDC’s urgent threats since 2013 [8,9].

With the scarcity of new antimicrobial agents in the pipeline, there is room for old antibiotics like colistin to perform against these MDR gram negative strains [10]. However, regarding colistin, the lack of relevant pharmacokinetic (PK) and pharmacodynamic (PD) studies leads to an ununiformed usage among countries. Due to the high prevalence of MDR *P. aeruginosa* and the potential nephrotoxicity of colistin [11,12,13,14], the scientific base of colistin instruction is worth being considered to improve the effectiveness and to reduce confusion in the clinical practice. It is especially important for critically ill patients who are not only in life-threatening conditions but also vulnerable to medication’s PK alteration and at a high risk of hospital-acquired infections by gram-negative bacteria such as MDR *P. aeruginosa*. Therefore, in this study, we aim to the combine the concept of the Monte Carlo simulation, the knowledge about PK and PD of colistin, and the local microbiological profile of *P. aeruginosa* to evaluate current dosing regimens and to propose appropriate ones for the treatment of *P. aeruginosa* infection in critically ill patients.

## 2. Materials and Methods

### 2.1. Microbiology

The microbiological data were obtained from a general hospital in Hochiminh city of Vietnam. The minimum inhibitory concentration (MIC) of colistin and the antimicrobial susceptibility test (AST) result of all *P. aeruginosa* isolates were extracted from May 2019 to March 2020. The tests were performed by the N240 Card/Vitek2-Biomerieux system. Six groups of antibiotics were used for the AST including antipseudomonal cephalosporin (ceftazidime, cefepime), antipseudomonal penicillin/beta-lactamase inhibitor (ticarcillin/avibactam, piperacillin/tazobactam), antipseudomonal carbapenem (imipenem, meropenem), aminoglycoside (gentamycin, amikacin), antipseudomonal fluoroquinolone (ciprofloxacin, levofloxacin), and polymyxin (colistin). MIC_50_ and MIC_90_ of colistin were then calculated from the MIC distribution.

As stated by European Committee on Antimicrobial susceptibility testing (EUCAST), the isolates were susceptible to colistin if the MIC was equal or less than 2 mcg/L [15]. However, it should be kept in mind that Clinical and Laboratory Standards Institute (CLSI) recently issued a new guidance on AST standards, in which colistin efficacy has been reviewed and reclassified as intermediate inhibition against *P. aeruginosa* at MIC of 2 mg/L or below [16].

### 2.2. Pharmacokinetic Parameters

According to current findings, concentrations of colistimethate sodium (CMS- colistin’ prodrug) and colistin fit to a two-compartment and a one-compartment pharmacokinetic model, respectively [17,18,19]. In this study, we adapted the published equations and population PK parameters related to critically ill patients from Nation et al. (Table 1 and Equations (1)–(3)) [19].
(1)$$\frac{\mathrm{dCMSc}}{\mathrm{dt}}=R-\mathrm{CLD}1\;\times \;\left(\frac{\mathrm{CMSc}}{V1}-\frac{\mathrm{CMSp}}{V2}\right)-\left(\mathrm{CLTCMS}\;\times \;\frac{\mathrm{CMSc}}{V1}\right).$$
(2)$$\frac{\mathrm{dCMSp}}{\mathrm{dt}}=\mathrm{CLD}1\;\times \;\left(\frac{\mathrm{CMSc}}{V1}-\frac{\mathrm{CMSp}}{V2}\right)$$
(3)$$\frac{\mathrm{dCol}}{\mathrm{dt}}=\mathrm{CLNRCMS}\;\times \;\frac{\mathrm{CMSc}}{V1}-\mathrm{CLTc}\;\times \;\frac{\mathrm{Col}}{V3}$$

### 2.3. PK/PD Index and PDT

The magnitude of the colistin-related PK/PD index, which is the area of free colistin concentration under the curve over MIC (*f*AUC/MIC), was extracted from the study of Cheah. et al. [20]. The target of *f*AUC/MIC chosen for this study is 10, which effectively produces 1 log kill against the MDR strain 19,056 in the mouse thigh infection model.

### 2.4. Monte Carlo Simulation

The Monte Carlo simulation was run by Crystal Ball ^®®^ software (Oracle Crystal Ball ^®®^ version 11.1.2.4.850) on 10,000 virtual patients. The uniform distribution was used for the protein-binding fraction and creatinine clearance while the log-normal distribution was applied to all other PK parameters [21,22,23]. The *f*AUC in 24 h (*f*AUC_0__-24_), probability of target attainment (PTA), and cumulative fraction of response (CFR) were computed from the free colistin concentration, MIC range, and MIC distribution. PTA of 80% and CFR of 90% were the targets of bacteria eradication in our study.

For each regimen, the trough concentration at steady state (C_trough,ss_) was compared with the values of 2.42 mg/L and 3.33 mg/L from the literature [24]. The patients presented with C_trough,ss_ above the cut-off value of 2.42 mg/L and 3.33 mg/L were considered to possibly have a higher risk of nephrotoxicity at the end of treatment (EOT) and at day 7 (D7), respectively.

The tested regimens in our study were from the current recommendation of Food and Drug Administration of United State (US-FDA), European Medicines Agency (EMA), Nation et al. [17,25,26], and our suggestions (Table 2). In each regimen, loading doses (LDs) and maintenance doses (MDs) were given in mg of colistin base activity (CBA) and were theoretically administered in 30 min.

Our study was approved by the ethical committee of the Faculty of Dentistry/Faculty of Pharmacy, Mahidol University (No: COA. No. Mu-DT/PY-IRB 2020/013.1702), and we received permission to use laboratory results from FV hospital (Hochiminh city, Vietnam).

## 3. Results

### 3.1. Microbiology

Of 137 datasets of *Pseudomonas aeruginosa* isolates from all sources (sputum, blood, urine, abscess pus, etc.), 16 sets were excluded (15 duplicates and 1 inadequate information). Tested antibiotics for AST were antipseudomonal cephalosporin (ceftazidime, cefepime), antipseudomonal penicillin/beta-lactamase inhibitor (ticarcillin/clavulanic, piperacillin/tazobactam), antipseudomonal carbapenem (imipenem, meropenem), aminoglycoside (gentamycin, amikacin), antipseudomonal fluoroquinolone (ciprofloxacin, levofloxacin), and polymyxin (colistin). According to the AST in Table 3, 24.8% of isolates (30/121) were MDR (not susceptible to at least 1 agent in at least 3 groups of antibiotics [27]). Among these, 70% of isolates were susceptible to only polymyxin. Colistin remained the most effective agent with only 0.8% of resistance. MIC_50/90_ of colistin was 0.5 mg/L (Figure 1).

### 3.2. PTA of All Regimens in Different Ranges of Creatinine Clearance (CrCl)

The PTA results of all regimens among various renal function ranges were depicted in Figure 2a–f. At MIC of 2 mg/L, none of the regimens achieved 80% PTA in the group of CrCl 101–130 mL/min. Even at the high dose of 600 mg, the PTA was only around 75%. For those patients with CrCl 80–100 mL/min, only 1 regimen from our study (120 mg every 6 h) was able to reach approximately 80% PTA. At CrCl of 51–79 mL/min, the PTA of approximately 80% and above was seen in three regimens with 360 mg per day. PTA of 80% and beyond was produced by regimens with total daily doses (TDDs) of at least 220 mg, 120 mg, and 80 mg in the three remaining groups of CrCl 30- 50 mL/min, 11–29 mL/min, and 1–10 mL/min, respectively. On the other hand, high PTA results (above 80%) were recorded at MIC of 1 mg/L and 0.5 mg/L with lower TDDs. Especially, at MIC of 0.5 mg/L, 80% PTA was found in all of the tested regimens. With MIC above 2 mg/L, none of the regimens were able to achieve the PTA target regardless of renal function.

Considerably, in some CrCl clusters, with the same TDD, regimens with different frequencies showed different PTAs. For instance, in the group of CrCl 101–130 mL/min, with the daily dose of 400 mg, the 100 mg every six-hour regimen produced higher PTA compared with the 200 mg every 12-h regimen. The difference was 2.02% (90.93% versus 88.91%) and 5.5% (60.55% versus 55.05%) at MIC of 0.5 mg/L and 2 mg/L, respectively. In the group of 80–100 mL/min CrCl, there was about a 3% difference between two regimens of 150 mg every 12 h and 100 mg every 8 h, at MIC of 2 mg/L. Or for those patients with CrCl 51–79 mL/min, the regimens of 180 mg every 12 h, 120 mg every 8 h, and 90 mg every 6 h produced disparate PTAs with 79.6%, 80.15%, and 84.87%, respectively. For the group of CrCl 1–10 mL/min, less distinguished PTA was found between two regimens, 80 mg every 24 h and 40 mg every 12 h (i.e., 84.16% and 85.99%, respectively).

### 3.3. CFR and Nephrotoxicity Risk

Both CFR and AKI risk estimation were displayed in Table 4 and Table 5. As stated in those tables, most regimens achieved CFR 90% and above. Only in the group of patients with very good renal function (100–130 mL/min) could we observe some CFR values from 80% to less than 90%. For all tested dosages, the proportion of patients related to higher AKI risk spread out widely from 5% to 90%. The chance of developing AKI at the end of treatment was higher compared to that on day 7. The US-FDA recommendation had almost a lower AKI risk than those from EMA and Nation et al.

### 3.4. The Recommendation of Our Study

On the basis of the PTA target of 80% and appropriate AKI risk, we made three colistin dosage recommendations according to CrCl ranges at MIC of 0.5 mg/L (Table 5), 1 mg/L (Table 6), and 2 mg/L (Table 7). With CFR of 90% and above, the dosing regimen at MIC of 0.5 mg/L, which was also the MIC_90_ of our hospital, could be considered as the empirical treatment of colistin in this hospital.

## 4. Discussion

In Vietnam, a multicenter observational study by Biedenbach et al. in 2016 reported the carbapenem-resistant *P. aeruginosa* (CRPA) rate of 66.5% and MIC_90_ of 0.5 mg/L [11]. According to our small-scale study, the MIC_90_ was still the same with 0.5 mg/L. However, the resistant rates were lower, with 24.8% and 28.9% for MDR *P. aeruginosa* and CRPA, respectively. The different rates of CRPA between studies could be attributed to the patients’ characteristics (underlying diseases), carbapenem use, and infection control [28]. The study of Biedenbach et al. focused on hospital-acquired and ventilator-associated pneumonia in five big tertiary hospitals across Vietnam, which suggested that they had many patients with complicated underlying diseases and ICU admission [29]. Meanwhile, our data were from all types of infections in a small general private hospital. The less severe of patients’ condition as well as the small number of patients were assumed to affect the lower usage of all antibiotics in general and carbapenem in specific. Of note, the low MIC_90_ and low colistin-resistant rate of *P. aeruginosa* in our study suggested the necessity of colistin against this organism in our population. Therefore, it is prudent to rationalize the dose to increase the efficacy as well as to reduce the resistant incident of colistin.

According to the PTA results, we found some relationship between patients’ renal function and PTA achievement. At the MIC breakpoint of 2 mg/L, PTA fell under the target (80%) in patients with normal renal function (101–130 mL/min). All regimens from US-FDA, EMA, and Nation (300 to 360 mg daily) achieved less than 50% PTA in this CrCl range. Meanwhile, most of those recommendations showed more than 80% PTA in patients with CrCl less than 50 mL/min. At the low MIC of 1 mg/L, the conventional regimens (from US-FDA, EMA, and Nation) similarly failed to gain 80% of PTAs in the group of CrCl 101–130 mL/min. This renal function-related effect was also seen in the study by Jitaree et al. [30]. Their study was about the efficacy of colistin against carbapenem-resistant *Klebsiella pneumoniae* and *Escherichia coli*. They reported the low PTA (65–89%) achieved in patients with CrCl > 50 mL/min at MIC ≥ 2 mg/L. This result also confirmed the study by Garonzik et al. [19]. Upon the observation on 115 critically ill patients, they found that patients with good renal function, who were given higher CMS dose, had lower colistin concentration than the others. These findings were supported by the current acknowledgment of CMS’ renal elimination that it was in parallel with CrCl [19,31]. At CrCl of 25 and 120 mL/min, the values of CLrcms were estimated at around 24 to 26 and 92 to 123 mL/min, respectively. Moreover, in a PK study of 73 patients, Gregoire et al. pointed out that the conversion fraction of CMS to colistin depended on renal function [31]. They predicted that at CrCl of 120 mL/min, only 32% of CMS transformed to colistin, but at 10 mL/min of CrCl, this rate increased by up to 80%. Consequently, at high kidney function, the formed colistin concentration was low. This fact, again, cast doubt on the efficacy of colistin against strains with MIC of 2 mg/L and for patients with good renal function.

Furthermore, the US-FDA recommendation was inferior to the other two popular regimens from EMA and Nation. In almost renal function groups, PTAs from US-FDA dosing regimens were lower than those from the other two regimens. The higher the MIC was, the bigger the PTA gap was found between regimens. For example, in the range of CrCl 51–79 mL/min, PTA magnitude from the US-FDA regimen was around 2% lower than the PTA from EMA and Nation at MIC of 0.5 mg/L. However, this difference was five times higher at MIC of 2 mg/L. The PTA difference was up to 50% in the group of CrCl 11–29 mL/min at MIC of 2 mg/L. We found the same results in the study by Jitaree et al. [30]. In which, EMA and Nation regimens showed more than 80% PTA for those with CrCl from 50 mL/min downward. While US-FDA recommendations continued showing much lower PTA. This inferiority was contributed to the low dose found in US-FDA regimens compared to others. Among these three regimens, only US-FDA used the weight-based dosing. When applying on a lower body weight population (60-kg body weight in both our study and Jitaree et al. versus 80-kg typical body weight), the lower doses and lower PTAs were the obvious consequences [32]. As stated in all studies by Garonzik et al., Nation et al., and Gregoire et al., patients’ body weight had little impact on the maintenance dose and was included only in the form of CrCl in the equations developed by Garonzik and Nation [17,19]. Therefore, a weight approach like US-FDA suggestions might result in poor performance, especially when being used for a population with a small body size such as that of the Vietnamese population.

Remarkably, despite the satisfied PTAs, both EMA and Nation recommendations demonstrated a high chance of AKI development in clusters of reduced renal function (CrCl ≤ 50 mL/min). Around 50% to 80% and 63% to 88% of patients had high colistin concentration-related AKI risk at day 7 and at the end of treatment. The lower the CrCl was, the higher the AKI risk was. Although no relationship between the colistin cumulative dose and nephrotoxicity has been found so far [33], those facts suggested the pile-up of colistin toxicity in patients with renal impairment. Forrest et al. [34] also reported that patients with lower kidney function (<80 mL/min) were more vulnerable to AKI, due to colistin exposure. Although many clinical studies observed that AKI happened quite early within 5 to 7 days of colistin initiation [35], our result, on the other hand, revealed that the chance of having AKI risk was higher at the end of treatment than this at day 7. Therefore, we suggested less aggressive approaches than those in EMA and Nation’s recommendation for patients with renal damage.

Given the poor performance of colistin in the upper ranges of CrCl, the questions were whether increasing colistin dose was appropriate and how high was enough. In our study, in the groups of CrCl higher than 50 mL/min, while the maximum approved dose from US-FDA, EMA, and Nation was from 230 to 360 mg CBA/day (EMA recommended 400 mg CBA in some cases), our tested doses were up to 600 mg CBA daily. More than 80% PTA was achieved with 480 mg CBA and 360 mg CBA daily for two groups of 80–100 mL/min and 51–79 mL/min, respectively. However, in the group of CrCl 101–130 mL/min, 74.72% was the maximum PTA achieved with 600 mg CBA daily. Similarly, Jitaree et al. required high doses of 450 to 540 mg CBA for 80% PTAs with CrCl ≥ 80 mL/min [30]. Considerably, the lack of a safety profile for these doses was the limitation. Moreover, at some point, the dose escalation was not directly proportional to the PTA increment. At MIC of 2 mg/L, when we increased the TDD by 33% (from 300 mg to 400 mg), the PTA raised 38.4%. However, when the dose was from 400 mg to 600 mg (50 %increase), only 23.4% of PTA increment was seen. Therefore, weighing between risk and benefit, the use of a high dose should be carefully considered together with the follow-up of renal function and antibiotic combination (if available).

An interesting finding in our study was the effect of the dosing interval. Theoretically, the regimens with the same daily dose would yield similar AUC. However, in our study, different PTAs were found in some regimens with various dosing intervals. For example, in the group of CrCl 101–130 mL/min, at MIC of 2 mg/mL around 5% PTA difference was seen between 100 mg every 6 h and 200 mg every 12 h. Or in the group of 80–100 mL/min, 3% higher PTA was displayed in 100 mg every 8 h compared to 150 mg every 12 h. Even 400 mg daily in a six-hour interval could produce more efficient PTA with 2% higher than that of 450 mg daily in an eight-hour interval. This difference was less visible along with the reduction of CrCl. Around 0.5% to 1.85% differences were seen in other ranges from 1 mL/min to 79 mL/min. Although observed by Jitaree et al. [30], this finding has been firstly reported in our study so far. In order to explain this discrepancy, we tried to look deeper into the PKPD characteristics of CMS and colistin. As a transformed product of CMS, basically, the formed colistin concentration would be increased when we used higher CMS dose. However, when comparing the rate constants k10 and k13 of the two processes of renal elimination and colistin conversion of CMS, respectively, we found that the elimination phase was predominant. From our simulation result, in the group of CrCl 101–130 mL/min, k10 (0.55 h^−1^, 0.05–8.6) was more than two times higher than k13 (0.22 h^−1^, 0.02–1.66). Therefore, when we increased the CMS dose, the ratio of CMS renal excretion was much higher than the ratio of colistin conversion. Furthermore, following the first-order kinetic, CMS concentration would be of a half reduction after one half-life time (i.e., 4.5 h). Therefore, the higher the concentration was, the more CMS would be eliminated. When we fractionated the daily dose into small parts with more frequent administration, we were trying to flatten the CMS curve with lower concentration. Consequently, the elimination amount was reduced and more formed colistin was produced. On the other hand, in the group of CrCl 11–29 mL/min, k10 and k13 were similar with 0.29 h^−1^ (0.03–1.6) and 0.23 h^−1^ (0.02–1.45), thus, the PTA difference was less significant. Secondly, according to Smith et al. [36], the ratio of half-life time (t_1__/2_) and interval (τ) impacted the steady-state concentration. The bigger the ratio of t_1/2_/τ was the higher the accumulation was. Hence, accumulated CMS would reduce according to the extension of the dosing interval (using the same TDD). Those reasons somehow made our findings logical and feasible. Notably, those differences were seen mostly in MIC range of 1 to 4 mg/L, where the most uncertainty of colistin efficacy could show up. At MIC ≤ 0.5 mg/L or ≥ 8 mg/L, at which PTA achievement was either too easy or too difficult, respectively, the distinguishment was barely seen. Therefore, utilizing the advantage of a dosing interval was another strategy to improve colistin performance around the MIC breakpoint (i.e., 2 mg/L) for patients with potent renal function.

On the basis of the results of the study, we proposed three dosing schemes at different MICs. The first one was at MIC of 0.5 mg/L which was also the MIC_50__/90_ of our population. At this point, all regimens were chosen according to CFR more than 90%. At MIC of 1 and 2 mg/L, our PTA target was 80% upward. At MIC of 2 mg/L, our study and many others proved that a high PTA was difficult to achieve [30,37]. Any effort to increase PTA might result in arising toxicities rather than benefits. Therefore, weighing the risk and benefit, a PTA target of 80% could be appropriate for MDR *P. aeruginosa* treatment. Furthermore, the nephrotoxicity rate of colistin was reported from 20–76% in clinical research [33,38,39]. However, this side effect is reversible when discontinuing colistin use. Therefore, balancing the toxicity risk and the severity condition in critically ill patients, at some point, we decided to use high dose colistin with a relatively high chance of AKI development.

When comparing with the recommendations from Jitaree et al., our suggestions were mostly lower. This contributed to the different pharmacodynamics targets. Jitaree et al. studied infection by *K. pneumoniae* and *E. coli* with a target of *f*AUC/MIC ≥ 25. While our target was MDR *P. aeruginosa* with *f*AUC/MIC ≥10. Additionally, our PTA target was also lower than the target in Jitaree et al. (80% versus 90%). This difference suggested that the colistin dose was not the same for various GNB treatments, and a common regimen was no longer appropriate. A low dose regimen if possible should be highly recommended for its benefit.

### Limitation of Our Study

Our study has some limitations. Firstly, we only assessed the target of 1 log kill from the mouse thigh infection model. Therefore, our result should not be interpreted in the context of lung infection. Secondly, the method to perform microbiologic results (Vitek 2) and the population (all types of infection and patient) might underestimate the real MIC in critically ill patients. Finally, all regimens based on CFR are very specific to a certain health-care setting. As a result, they should not be used widely without considering the various MIC distributions.

## 5. Conclusions

Our study was successful in comparing the conventional dosing regimens and our own suggestion of colistin against *P. aeruginosa* in our local microbiological data of the critically ill patient population. The use of fractionated colistin regimens with high doses (and combination therapy, if possible) was recommended, especially for patients with competent renal function. For patients with low colistin MIC or reduced kidney function, less aggressive strategies should be considered to balance the risks and benefits. With further investigation, our regimen at MIC of 0.5 mg/L (MIC_90_) could be adapted as the empiric therapy for *P. aeruginosa* treatment (except for the lung infection) in critically ill patients in the studied hospital to avoid unnecessarily high doses.

## Figures and Tables

**Figure 1 antibiotics-10-00595-f001:**
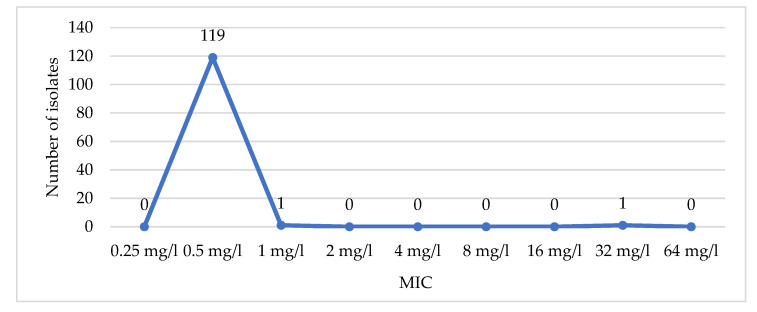
Minimum inhibitory concentration (MIC) distribution of colistin among *P. aeruginosa* isolates.

**Figure 2 antibiotics-10-00595-f002:**
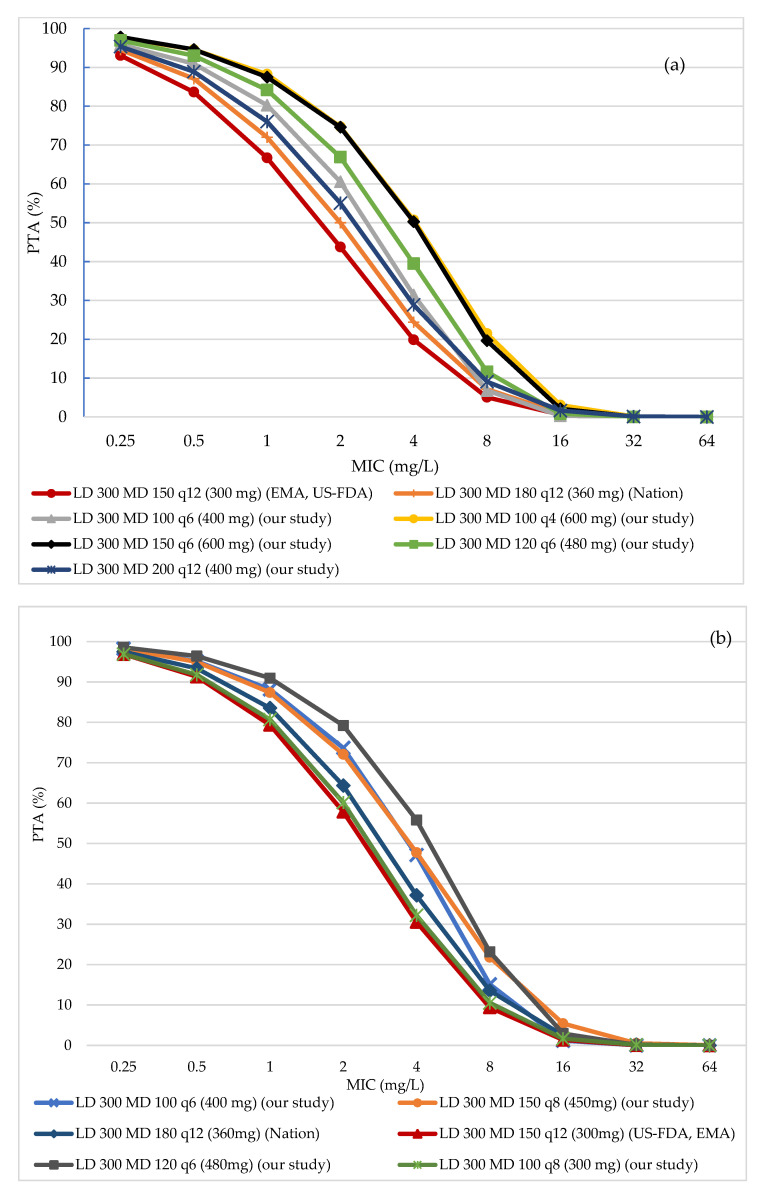
Probability of target attainment (PTA) achievement in the groups of CrCl 101–130 mL/min (**a**), 80–100 mL/min (**b**), 51–19 mL/min (**c**), 30–50 mL/min (**d**), 11–29 mL/min (**e**), 1–10 mL/min (**f**). The detail of PTA results is found in Appendix A. LD: loading dose (in mg colistin base activity), MD: maintenance dose (in colistin base activity), US-FDA: Food and Drug Administration of United State, EMA: European Medicines Agency, q12: every 12 h.

**Table 1 antibiotics-10-00595-t001:** The population pharmacokinetic (PK) parameters of colistin in critically ill patients.

Parameters	CMS (SD)	Parameters	Colistin (SD)
V1 (L)	12.9 (5.2116)	V3/fm (L)	57.2 (24.882)
V2 (L)	16.1 (11.4149)	CLRCsl-pop/fm (L/h/CrCl)	0.00834
CLD1 (L/h)	9.57 (7.66557)	CLNRc/fm (L/h)	3.11
CLRslope (L/h/CrCl)	0.034 (0.02557)	fu	0.49 (0.11)
CLNRcms (L/h)	2.52 (1.00296)		

V1: central volume distribution of CMS, V2: peripheral volume distribution of CMS, CLD1: distributional clearance between central and peripheral compartments of CMS, CLRslope: renal clearance of CMS, CLnrcms: non renal clearance of CMS, V3/fm: apparent volume distribution of colistin, CLRCsl-pop/fm: apparent renal clearance of colistin, CLRNc/fm: apparent renal clearance of colistin, fu: unbound fraction of colistin, CMSc: central concentration of CMS, CMSp: peripheral concentration of CMS, R:, V1: central volume distribution of CMS, V2: peripheral volume distribution of CMS, Col: colistin concentration, CLTCMS: total clearance of CMS, CLNRCMS: non renal clearance of CMS, CLTc: total clearance of colistin, V3: volume distribution of colistin.

**Table 2 antibiotics-10-00595-t002:** Tested dosing regimens.

Renal Function ^a^	LD ^b^	MD ^b^	Regimen
101–130 mL/min	300 mg	150 mg Q12h	US-FDA, EMA
		180 mg Q12h	Nation
		100 mg Q6h	Proposed
		120 mg Q6h	Proposed
		200 mg Q12h	Proposed
		100 mg Q4h	Proposed
		150 mg Q6h	Proposed
80–100 mL/min	300 mg	150 mg Q12h	US-FDA, EMA
		180 mg Q12h	Nation
		150 mg Q8h	Proposed
		100 mg Q6h	Proposed
		120 mg Q6h	Proposed
		100 mg Q8h	Proposed
51–79 mL/min	300 mg	150 mg Q12h	EMA, Nation
		115 mg Q12h	US-FDA
		180 mg Q12h	Proposed
		120 mg Q8h	Proposed
		100 mg Q8h 100 mg Q12h	Proposed Proposed
30–50 mL/min	300 mg	75 mg Q12h	US-FDA
		125 mg Q12h	EMA
		110 mg Q12h	Nation
		250 mg Q24h 200 mg Q24h 100 mg Q12h	Proposed Proposed Proposed
		80 mg Q8h	Proposed
		120 mg Q12h	Proposed
		100 mg Q24h	Proposed
11–29 mL/min	300 mg	90 mg Q36h	US-FDA
		90 mg Q12h	EMA
		80 mg Q12h	Nation
		150 mg Q24h	Proposed
		200 mg Q36h	Proposed
		120 mg Q24h 100 mg Q24h	Proposed Proposed
		60 mg Q12h	Proposed
		66 mg Q24h	Proposed
1–10 mL/min	300 mg	60 mg Q12h	EMA
		70 mg Q12h	Nation
		80 mg Q24h 66 mg Q24h	Proposed Proposed
		33 mg Q24h	Proposed
		40 mg Q12h	Proposed

^a^: renal function is stratified by creatinine clearance in mL/min, ^b^: colistin dose is expressed as mg of colistin base activity. LD: loading dose, MD: maintenance dose, US-FDA: Food and Drug Administration of United State, EMA: European Medicines Agency, Q: every.

**Table 3 antibiotics-10-00595-t003:** Antimicrobial susceptibility test (AST) results of all *P. aeruginosa* isolates.

Total Included Isolates (*n* = 121)	Number (*n*)	Percentage (%)
Source of samples		
Blood	5	4.1
Urine	22	18.2
Respiratory tract	46	38
Others (skin, body fluid, etc.)	48	39.7
Resistance pattern (antibiogram)		
No resistance	38	31.4
Resistant to Ceftazidime	25	20.7
Resistant to Cefepime	27	22.3
Resistant to Ticarcilline/Clavulanic	68	56.2
Resistant to Piperacillin/Tazobactam	35	28.9
Resistant to Carbapenem	35	28.9
Resistant to Aminoglycoside	30	24.8
Resistant to Flouroquinolone	39	32.2
Resistant to Colistin	1	0.8
Multi-drug resistant	30	24.8

**Table 4 antibiotics-10-00595-t004:** PTA, CFR and AKI risk of simulated regimens from US-FDA, EMA, Nation, and our study.

CrCl (mL/min)	Regimens (TDDs*)		PTA (%)		CFR (%)	AKI Risk (%)	
			MIC 0.5 mg/L (MIC_90_)	MIC 2 mg/L		EOT	D7
101–130	150 mg Q12h (300 mg)	US-FDA, EMA	83.65	43.75	82.82	24.63	16.75
	180 mg Q12h (360 mg)	Nation	87.1	49.99	86.26	29.23	20.17
	200 mg Q12h (400 mg)	Our study	88.91	55.05	88.07	33.01	23.14
	100 mg Q6h (400 mg)	Our study	90.93	60.55	90.09	43.38	30.33
	120 mg Q6h (480 mg)	Our study	93.68	63.27	92.82	51.31	38.08
	100 mg Q4h (600 mg)	Our study	94.6	74.72	93.77	62.77	50.51
	150 mg Q6h (600 mg)	Our study	96.62	74.6	93.78	60.65	48.24
80–100	150 mg Q12h (300 mg)	US-FDA, EMA	91.32	57.85	90.47	36.59	26.24
	180 mg Q12h (360 mg)	Nation	93.37	64.36	92.52	42.58	31.83
	150 mg Q8h (450 mg)	Our study	94.96	72.07	94.12	56.45	45.01
	100 mg Q6h (400 mg)	Our study	95.12	73.69	94.28	59.45	46.34
	120 mg Q6h (480 mg)	Our study	96.42	79.23	95.58	66.74	54.62
	100 mg Q8h (300 mg)	Our study	91.79	60.15	90.94	42.57	30.66
51–79	115 mg Q12h (230 mg)	US-FDA	94.38	64.52	93.53	42.93	30.9
	150 mg Q12h (300 mg)	EMA, Nation	96.52	75.02	95.67	54	42
	180 mg Q12h (360 mg)	Our study	97.21	79.6	96.37	59.64	48.22
	120 mg Q8h (360 mg)	Our study	97.3	80.15	96.46	65.75	54.22
	100 mg Q8h (300 mg)	Our study	96.72	75.88	95.87	59.81	46.89
	100 mg Q12h (200 mg)	Our study	93.7	59.61	92.84	38.44	26
30–50	75 mg Q12h (150 mg)	US-FDA	97.66	69.6	96.79	48.2	34.7
	125 mg Q12h (250 mg)	EMA	99.18	86.35	98.34	68.73	56.18
	110 mg Q12h (220 mg))	Nation	98.71	83.11	97.86	63.45	50.74
	120 mg Q12h (240 mg)	Our study	99.07	84.77	98.23	67.98	54.81
	80 mg Q8h (240 mg)	Our study	98.95	86.15	98.11	73.17	61.38
	100 mg Q12h (200 mg)	Our study	98.48	79.97	97.63	60.15	46.86
	200 mg Q24h (200 mg)	Our study	98.62	79.26	97.77	41.24	30.2
	100 mg Q24h (100 mg)	Our study	93.77	47.3	92.87	19.07	11.46
	250 mg Q24h (250 mg)	Our study	99.18	85.6	98.34	47.49	37.15
11–29	90 mg Q36h ((60 mg)	US-FDA	97.01	39.17	96.06	11.51	5.79
	90 mg Q12h (180 mg)	EMA	99.77	92.21	98.94	78.29	66.68
	80 mg Q12h (160 mg)	Nation	99.71	90.59	98.88	74.87	62.55
	120 mg Q24h (120 mg)	Our study	99.39	81.28	98.54	45.21	33.16
	60 mg Q12h (120 mg)	Our study	99.54	82.8	98.69	63.74	48.63
	66 mg Q24h (66 mg)	Our study	97.33	52.72	96.42	23.17	13.08
	100 mg Q24h (100 mg)	Our study	98.98	74.37	98.11	39.51	26.54
	150 mg Q24h (150 mg)	Our study	99.76	88.35	98.92	53.79	41.79
	200 mg Q36h (133 mg)	Our study	99.78	83.31	98.94	32.3	23.16
1–10	60 mg Q12h (120 mg)	EMA	99.96	95.24	99.14	83.68	72.27
	70 mg Q12h (140 mg)	Nation	99.97	95	99.15	87.61	78.67
	80 mg Q24h (80 mg)	Our study	99.85	84.16	99	52.76	39.14
	40 mg Q12h (80 mg)	Our study	99.77	85.99	98.93	68.95	53.11
	66 mg Q24h (66 mg)	Our study	99.6	76.33	98.74	45.19	32.46
	33 mg Q24h (33 mg)	Our study	95.6	40.12	94.65	17.79	9.52

TDDs*: total daily doses in mg of colistin base activity, Q: every, PTA: probability of target attainment, CFR: cumulative fraction of response, AKI: acute kidney injury, CrCl: creatinine clearance, MIC_90_: minimum inhibitory concentration required to inhibit the growth of 90% of organisms, EOT: end of treatment, D7: at day 7, US-FDA: Food and Drug Administration of United State, EMA: European Medicines Agency.

**Table 5 antibiotics-10-00595-t005:** Recommendation for MIC of 0.5 mg/L (MIC_90_ of our hospital).

CrCl (mL/min)	Regimens (TDDs*)	PTA (%)	CFR (%)	AKI Risk (%)		Alternative Regimens (TDDs*)	PTA (%)	CFR (%)	AKI Risk (%)	
				EOT	D7				EOT	D7
101–130	100 mg Q6h (400 mg)	90.93	90.09	43.38	30.33					
80–100	100 mg Q8h (300 mg)	91.79	90.94	42.57	30.66	150 mg Q12h (300 mg) (US-FDA)	91.32	90.47	36.59	26.24
51–79	100 mg Q12h (200 mg)	93.7	92.84	38.44	26					
30–50	100 mg Q24h (100 mg)	93.77	92.87	19.07	11.46					
11–29	66 mg Q24h (66 mg)	97.33	96.42	23.17	13.08					
1–10	33 mg Q24h (33 mg)	95.6	94.65	17.79	9.52					

TDDs*: total daily doses in mg of colistin base activity, Q: every, PTA: probability of target attainment, CFR: cumulative fraction of response, AKI: acute kidney injury, CrCl: creatinine clearance, MIC_90_: minimum inhibitory concentration required to inhibit the growth of 90% of organisms, EOT: end of treatment, D7: at day 7, US-FDA: Food and Drug Administration of United State.

**Table 6 antibiotics-10-00595-t006:** Recommendation for MIC of 1 mg/L.

CrCl (mL/min)	Regimens (TDDs)	PTA (%)	AKI Risk (%)		Alternative Regimens (TDDs)	PTA (%)	AKI Risk (%)	
			EOT	D7			EOT	D7
101–130	120 mg Q6h (480 mg)	84.19	43.09	29.18				
80–100	100 mg Q6 (400 mg)	88.16	59.45	46.34				
51–79	100 mg Q8h (300 mg)	90.4	59.81	46.89	150 mg Q12h (300 mg)	89.99	54	42
30–50	100 mg Q12 (200 mg)	94.35	60.15	46.86	200 mg Q24h (200 mg)	94.08	41.24	30.2
11–29	100 mg Q24 (100 mg)	93.85	39.51	26.54				
1–10	66 mg Q24h (66 mg)	95.6	45.19	32.46				

TDDs*: total daily doses in mg of colistin base activity, Q: every, PTA: probability of target attainment, CFR: cumulative fraction of response, AKI: acute kidney injury, CrCl: creatinine clearance, MIC: minimum inhibitory concentration, EOT: end of treatment, D7: at day 7.

**Table 7 antibiotics-10-00595-t007:** Recommendation for MIC of 2 mg/L.

CrCl (mL/min)	Regimens (TDDs)	PTA (%)	AKI Risk (%)		Alternative Regimens (TDDs)	PTA (%)	AKI Risk (%)	
			EOT	D7			EOT	D7
101–130	Not recommend							
80–100	120 mg Q6h (480 mg)	79.23	66.74	54.62				
51–79	120 mg Q8h (360 mg)	80.15	65.75	54.22	180 mg Q12h (360 mg)	79.6	59.64	48.22
30–50	120 mg Q12h (240 mg)	84.77	67.98	54.81	250 mg Q24h (250 mg)	85.6	47.49	37.15
11–29	120 mg Q24h (120 mg)	81.28	45.21	33.16				
1–10	80 mg Q24h (80 mg)	84.16	52.76	39.14				

TDDs*: total daily doses in mg of colistin base activity, Q: every, PTA: probability of target attainment, CFR: cumulative fraction of response, AKI: acute kidney injury, CrCl: creatinine clearance, MIC: minimum inhibitory concentration, EOT: end of treatment, D7: at day 7.

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
