# Peer review of "Colistin Dosing Regimens against Pseudomonas aeruginosa in Critically Ill Patients: An Application of Monte Carlo Simulation"

_antibiotics, 2021, doi:10.3390/antibiotics10050595_

Round 1

Reviewer 1 Report

The manuscript provides a good add-on to the PK/PD of colistin for the treatment of P. aeruginosa. Overall, the manuscript is nicely presented. The data presentation on tables can be improved to make it easier for readers to get the key message.

Page 2, lines 51 – 57: Vitek2 system was used to evaluate the colistin susceptibility. I think it is important to note that this system is not the gold standard method and is suboptimal.

Page 5, line 119: The authors mentioned no resistance in 38 isolates. The authors can be more specific by stating the antibiotics tested.

 Pages 8 – 9: Table 4 and 5 can be combined to enable easier comparison with the regimens used in the study with the recommended regimens.

Page 11, line 207: Nomenclature error. Should be Klebsiella pneumoniae instead of Klebsiella pneumonia.

Author Response

Thank you very much for your comments. Please see the response in the attachment.

Reviewer 2 Report

Authors did a nice study concering the effectiveness of colistin against MDR gram negative strains.

My main ethical concern is in Conclusion. Just before conclusion authors state:

"Limitation of our study
...Therefore, our result should not be interpreted in the context of lung infection..."

This should be clearly stated in the Abstract and in the Conclusion of the paper.

Additional points should be addressed prior to possible publication:

Authors used uniform distribution and log-normal:

"The uniform distribution was used for 87 protein- binding fraction and creatinine clearance while the log- normal distribution was 88 applied to all other PK parameters."

Explanation (and citation) about these distributions should be included in the paper. Would results change if other types of distribution were used?

Monte Carlo simulation should converge in the results. To confirm this authors should perform several studies with variable number of virtual patients and show the convergence of the parameters. Only one type of the study is not enough.

In "3.4 The recommendation of our study"

authors state that colistin dosage recommendations were based on the PTA and AKI. But AKI is not presented in the Table 6 and should be added.

The AKI values are in agreement with the PTA for all cases in Table 6 but for the case (101-130). Clear explanation why the valued of 100 mgQ6h was chosen should be added.

In Fig. 12a units are missing in parenthesis. Moreover, Fig. 12 would be better represented as a 2D surface improving the visibility and readability of the Figure.

Author Response

Thank you very much for your review. Please see the response in the attachment.

Round 2

Reviewer 2 Report

No additional comments.